# Psychological Stress Reduces the Effectiveness of Periodontal Treatment: A Systematic Review

**DOI:** 10.3390/jcm14051680

**Published:** 2025-03-01

**Authors:** Kelly Rocio V. Villafuerte, Luiz Henrique Palucci Vieira, Karina O. Santos, Edgard Rivero-Contreras, Alan Grupioni Lourenço, Ana Carolina F. Motta

**Affiliations:** 1Grupo de Investigación Salud Integral Humana (GISIH), Faculty of Health Sciences, Universidad César Vallejo (UCV), Callao Campus, Lima 07001, Peru; 2Department of Biological Sciences, Bauru School of Dentistry, University of São Paulo (USP), Bauru 17012-901, SP, Brazil; karina_oliveira@usp.br; 3Department of Basic and Oral Biology, Ribeirão Preto School of Dentistry, University of São Paulo (USP), Ribeirão Preto 14040-900, SP, Brazil; lourenco@forp.usp.br; 4Department of Stomatology, Public Health and Forensic Dentistry, Ribeirão Preto School of Dentistry, University of São Paulo (USP), Ribeirão Preto 14040-904, SP, Brazil; anacfm@usp.br

**Keywords:** psychological stress, periodontitis, non-surgical periodontal treatment, periodontal debridement, systematic review

## Abstract

**Background/Objectives**: To systematically evaluate scientific evidence related to the influence of psychological stress on the response to periodontal treatment. **Methods**: PubMed/NCBI (National Center for Biotechnology Information, US National Library of Medicine), Web of Science (ClarivateTM), EBSCOHost, SCOPUS, and ProQuest databases were searched for published clinical studies in English up to May 2024. The quality of each study was assessed using the Ottawa–Newcastle scale. **Results**: Of 803 relevant articles identified, 8 were included in the qualitative synthesis qualitative synthesis. These studies involved 445 patients who completed the follow-up period, ranging from 6 weeks to 6 months. Stressed patients were more likely to experience higher levels of PPD and BOP compared to non-stressed patients. In total, 75% of the included studies showed a positive relationship between stress and response to NSPT, 12.5% observed a negative relationship, and the remaining 12.5% found some degree of relationship in the results of clinical periodontal parameters. The level of evidence is categorized according to the quality of the synthesis presented. **Conclusions**: There is a positive correlation between psychological stress and periodontal treatment response, indicating that stress may negatively influence the clinical outcomes of NSPT. Stress may reduce the inflammatory response, which is crucial for eliminating periodontal micropathogens after periodontal treatment.

## 1. Introduction

Psychological stress is defined as an organism’s response to external or internal demands perceived as challenging or threatening, which cause physical or mental tension that destabilizes the current human equilibrium [1]. It can be classified as acute or chronic stress [2], which can temporarily activate innate immunity or suppress immune function in the case of chronic or prolonged stress, thus compromising the body’s ability to control infections or tissue repair processes [3], and this effect is related to the continuous activation of the hypothalamic–pituitary–adrenal (HPA) axis, which deregulates inflammatory responses capable of damaging the individual’s general state of health [2,4].

Studies that have explored the relationship between psychological stress and periodontitis have indicated that stress is associated with various systemic diseases, such as diabetes mellitus and autoimmune diseases, which contribute to susceptibility to infections, including periodontitis [5,6]. Periodontitis is an inflammatory disease of infectious origin caused by pathogenic bacteria that is influenced by a combination of factors, including risk factors such as diabetes and smoking [7]. In addition, stress can lead individuals to adopt unhealthy habits, such as a poor diet and a greater propensity to abuse alcohol and cigarettes, contributing to the worsening of periodontitis [5,7]. Socioeconomic factors such as low schooling and lack of dental access can also increase exposure to chronic stress and reduce an individual’s ability to maintain good oral health [8].

Research has provided evidence on the influence of psychosocial factors on the development and progression of periodontitis [3,9], for example, Genco et al., 1998 [9] in a cohort study, which analyzed the relationship between stress and periodontal disease in 1426 patients, denoted that stress can alter the host’s immune response and promote gingival inflammation. On the other hand, Decker et al., 2021 [3], in a meta-analysis of 12 studies with a total of 2367 participants, showed that prolonged psychological stress can modulate the expression of pro-inflammatory cytokines, negatively affecting the host’s immune response.

Psychological stress can threaten periodontal immune homeostasis in several ways, firstly through its influence on the neuroendocrine system, affecting the HPA axis, the sympathetic autonomic nervous system, and the peptidergic nervous system [7]. This can lead to immune suppression, with a consequent increase in the production of glucocorticoids and pro-inflammatory cytokines, such as interleukin-1 (IL)-1 and tumor necrosis factor-alpha (TNF) [10], as well as causing a lower expression of anti-inflammatory cytokines, which can exacerbate gingival inflammation and the destruction of periodontal tissues, favoring the development of periodontitis [7,11]. These complex mechanisms describe how psychological stress can directly and indirectly influence periodontal health [7].

In addition, longitudinal clinical cohort studies [12,13,14,15] have shown that patients with high psychological stress and inadequate coping behavior had greater attachment loss and alveolar bone loss compared to patients who managed their stress better. Furthermore, in other medical fields such as oncology, a systematic review [16] indicates that poor psychosocial adjustment and inadequate coping strategies increase the risk of cancer recurrence. These findings highlight the importance of assessing mental health during diagnosis to optimize treatment and patient well-being.

In the periodontal context, although non-surgical periodontal therapy (NSPT) is often effective in the treatment of periodontitis, stress-related factors can also negatively affect outcomes [17] and therefore clinical studies [18,19] have emerged indicating that psychological stress can negatively influence the response to periodontal treatment. Likewise, a recent cross-sectional study [20] of 203 patients highlights the importance of assessing psychological state in dental treatment, as it found a significant association between perceived stress and periodontal health.

Understanding the influence of psychological stress on periodontal health is fundamental for developing more effective prevention and treatment strategies. Considering the emerging evidence from previous systematic reviews indicating the relationship between psychological stress and periodontal health [21,22,23,24,25], or psychological stress as a risk factor and progression of periodontitis [22,23,25], and whether periodontitis has a significant effect on salivary cortisol [21,24], reflecting changes in free cortisol levels in the blood [24]. However, despite the existence of a review protocol (i.e., justification and methods), to date, no results have been collected specifically on the effect of psychological stress on periodontal treatment outcomes. This reinforces the need for our systematic review. Therefore, this systematic review aims to analyze the influence of stress on the response to periodontal treatment.

## 2. Materials and Methods

Study Registration: The present systematic review was structured according to the Preferred Reporting Items for Systematic Reviews and Meta-Analyses (PRISMA) guidelines. This protocol was registered in the Open Science Framework (OSF) Generalized Systematic Review Registration (ID: qfs6p) and is also available in full elsewhere [26]. The patient, exposure, comparison, outcome (PECO) strategy was used to formulate the research question: Does psychological stress influence the response to periodontal treatment?

### 2.1. Eligibility Criteria

Type of studies to be included: case–control studies; cohort or cross-sectional; published in English up to May 2024 that examined the influence of psychological stress on periodontal clinical outcomes, and using the following PECO strategy:

*Participants/population*: Adult patients diagnosed with periodontal disease and who will receive periodontal treatment (Subgengival scaling and root planing or Subgengival mechanical debridement).

*Exposure(s)*: Psychological stress [assessed by psychometric instruments—questionnaires and/or biomarkers to categorize stress (e.g., cortisol)].

*Comparison (C):* Patients without psychological stress [assessed by psychometric instruments—questionnaires and/or biomarkers to categorize stress (e.g., cortisol)] and assessed after periodontal therapy.

*Outcome measures (O)*: Change in periodontal clinical measures after periodontal treatment.

Primary outcomes consisted of changes in probing pocket depth (PPD), clinical attachment level (CAL), and bleeding on probing (BOP), and secondary outcomes consisted of changes in plaque index (PI).

Exclusion criteria: (1) Observational studies that did not perform periodontal therapy; (2) studies performing periodontal therapy together with other adjunctive therapies (e.g., probiotics, symbiotics, photodynamic therapy, antibiotics, or psychotropic medications); (3) or any other periodontal treatment in the previous 6 months; (4) patients with systemic or autoimmune diseases, which influence treatment outcomes (e.g., diabetes); (5) pregnant women; (6) patients using immunosuppressive drugs or drugs that affect the oral microbiome (example: antineoplastic drugs); (7) studies that did not pass through an ethics committee; (8) letters, case reports, brief communications; (9) studies in animal models and in vitro.

Information sources and search strategy: The electronic database search included articles indexed in MEDLINE through PubMed/NCBI (National Library of Medicine/U.S. Department of Health and Human Services), Web of Science (Clarivate™), EBSCOHost (EBSCO Information Services), SCOPUS (Elsevier), and ProQuest (ProQuest LLC). We searched for articles published in English until 31 May 2024.

The database search strategy included MeSH search descriptors; psychological stress, emotional stress, chronic stress, job-related stress mixed with Boolean operators OR, AND: Periodont* OR periodontal therapy OR non-surgical periodontal therapy OR scaling OR periodontal debridement OR mechanical debridement OR plaque removal AND control* OR baseline OR *stressed group* AND probing depth OR clinical attachment level OR bleeding of probing.

Study selection: Once the search step was completed, references were managed and duplicates were automatically removed in EndNote software (EndNote X7.0.1, Thomson Reuters©, New York, NY, USA), which facilitated reference management. Grey literature was not considered. The retrieved articles were exported to the Rayyan reference manager (https://www.rayyan.ai; last accessed on 5 September 2024). The selection process was conducted in two phases. Two reviewers (K.R.V.V. and K.O.S) independently screened the titles and abstracts of all retrieved references for relevance (Phase One). In the second phase, the articles selected in the previous stage were carefully evaluated and the full texts were judged. When disagreement was reached, a third reviewer (E.R) was consulted. Studies that met the inclusion criteria were subjected to validation and data extraction. Inter-reviewer reliability in the study selection process was determined by Cohen’s κ test, assuming an acceptable threshold value of 0.8 [27].

### 2.2. Data Extraction

Relevant data extracted from each study were: Investigators and year of publication; country; study design; participant characteristics, including diagnosis of periodontitis, psychological stress (psychometric instruments—questionnaires or biomarkers used), study inclusion and exclusion criteria; clinical parameter data (PPD, CAL, BOP, PI); outcome measures of interest to the review, main results, authors’ conclusions and source of funding (Table 1, Table 2 and Table 3 and Appendix A).

### 2.3. Methodological Quality Assessment

Two independent authors (KRVV and L.H.P.V) assessed the methodological quality of each included study using the Newcastle–Ottawa scale (NOS) [34], this scale is specifically designed for non-randomized studies.

The Newcastle–Ottawa Scale (NOS) uses a “star” system to obtain the quality score for each study, which is determined by 3 main components: (1) quality of study participant selection, (2) comparability, (3) exposure and outcome, with a maximum score of 9 points for case–control and cohort studies, while for cross-sectional studies the maximum score is 10 points. Methodological quality is divided into 3 categories: (1) high quality (total score: 7 to 9/10); (2) moderate quality (total score: 4 to 6); and (3) low quality (total score: 0 to 3). Disagreements were resolved through bilateral discussions (Table 4 and Table 5).

### 2.4. Evidence Synthesis

To present a synthesis of the results of the studies, the level of evidence was categorized as: (a) strong [consistent results, with ≥75% of the studies indicating results in the same direction, from several high-quality studies]; (b) moderate [consistent results observed in several moderate-quality studies and/or in a single high-quality study]; and (c) limited [consistent results identified in a single moderate-quality study and/or exclusively in low-quality studies].

### 2.5. Search Results and Excluded Trials

A total of 803 articles were identified through the database search strategy. Of these, 375 were excluded due to duplication. In total, 428 studies were reviewed. After the analysis of titles and abstracts, 418 were excluded. The full texts of the remaining 10 publications deemed eligible were reviewed. Finally, eight studies were included in the qualitative synthesis [18,19,28,29,30,31,32,33]. Two studies [12,35] were excluded because they did not meet the inclusion criteria (Figure 1).

Of note, all the included studies in the present review were conducted in duly registered institutions, such as universities and hospitals.

## 3. Results

### 3.1. Characteristics of the Included Studies

The characteristics of the included studies are shown in Table 1 and Appendix A. Of the eight included studies, one was case–control [28], and seven were cohort studies [18,19,29,30,31,32,33]. All had a parallel design and were published between 2005 and 2023. Two studies were conducted in India, two in France, two in Italy, one in Brazil, and one in the United Kingdom. A total of 467 patients initially participated (427 were diagnosed with periodontitis, 10 with gingivitis, and 30 without periodontitis). Of these, 445 patients (405 with periodontitis, 10 with gingivitis, and 30 without periodontitis) completed the follow-up period, which ranged from 3 to 6 months in seven studies [18,19,28,29,30,31,32], while in one study [33] the follow-up was 6 weeks. This resulted in a completion rate of 95.3%. In addition, the mean age of participants ranged from 28.8 to 55.2 years.

In five studies [18,28,30,31,33], smoking was an exclusion criterion. Three [18,28,33] of these five studies excluded heavy smokers (those consuming more than 10 cigarettes per day). In one study [29], the smoking status of the participants was recorded, while the remaining two studies did not specify or mention whether smokers would be included [19,32].

### 3.2. Periodontal Diagnosis

Four studies [18,19,32,33] employed the criteria proposed by the 2018 International Workshop on Periodontology [36], which diagnoses periodontitis in stages and grades. Another four studies [28,29,30,31] used an older criterion described by Armitage (1999) [37] to diagnose periodontitis. In addition, it is worth mentioning that before initiating periodontal therapy, Vettore et al. [28] classified pocket depth to probing into shallow PPD ≤ 4 mm, moderate (PPD values 4–6 mm), and deep pockets (PPD > 6 mm), Bakri et al. [29] classified periodontal pockets with bleeding and without bleeding, Varadhan et al. [30] included patients with CAL loss ≥ 3 mm and PPD > 5 mm at multiple sites in all four quadrants of the mouth. Additionally, a study [31] included patients with gingivitis and periodontitis with PPD between 4 and 6 mm and CAL of 3–4 mm.

Seven studies [18,19,28,29,30,32,33] used hand-held probes to record measurements of periodontal parameters before and after periodontal therapy. One study [31] did not mention which instrument was used. Five studies [18,19,28,30,33] used the University of North Carolina probe (PCP UNC 15, Hu-Friedy, Chicago, IL, USA), while two studies [29,32] used the Williams periodontal probe. In addition, all seven studies mentioned that intra- and interexaminer calibration of the evaluator was performed before the start of the study.

### 3.3. Stress Diagnosis (Psychological Measures and Biomarkers)

Of the eight studies included, four studies [18,29,32,33] used the Perceived Stress Questionnaire (PSS-10), one study [28] used the Stress Symptom Inventory (SSI), another study [19] opted for the Depression, Anxiety, and Stress Scale 42 (DASS-42), Ratika et al. [31] used the General Health Questionnaire (GHQ), and finally Varadhan et al. [30] used the Derogatis Stress Profile questionnaire (DSP). The study by Dubar et al. [32] in addition to assessing stress with PSS-10, used the STAI-Y questionnaire to assess anxiety. Likewise, three studies [18,19,33] used stress-coping questionnaires such as the Toulouse Coping Scale (TSC) [19], the Stress-Related Vulnerability Scale (SVS) [33], and the Coping Response Inventory (CRI) [18], to understand the coping style of participants in stressful or challenging situations.

As for biological biomarkers, four studies [19,29,30,32] assessed cortisol levels for diagnosing psychological stress: three in unstimulated saliva [29,30,32] and one study in plasma [19]. Two studies [29,32] used the enzyme-linked immunosorbent assay (ELISA) method to measure cortisol levels in ng/mL, while one study [30] employed chemiluminescence immunoassay (CLIA) using microplatinometers, which have a sensitivity of less than 0.5 µg/mL, and the study [19] that performed plasma cortisol measurements used a cobas and 601 automated immunoassay kit. In addition, two studies [19,31] evaluated chromogranin-A (CgA) levels in response to psychological stress, in one study it was measured by an automated immunoassay kit (Kryptor Compact Plus, Thermo Fisher, Illkirch-Graffenstaden, France) [19] and by enzyme-linked immunosorbent assay (ELISA) [31].

Bakri et al. [29] also evaluated biomarkers of elastase (ng/mL) and C-terminal Teleopeptide C of collagen type 1 (ICTP/ng/mL) in the gingival crevicular fluid (GCF) to evaluate collagen type 1 degradation and bone loss caused by periodontitis. To detect elastase activity was measured by a modification of the method of Giannopoulou (1992) [38], and for ICTP detection, an enzyme-linked immunosorbent assay ELISA was used following the manufacturer’s instructions. The authors observed that elastase levels in GCF were statistically higher in stressed patients.

### 3.4. Non-Surgical Periodontal Therapy (NSPT)

All included studies [18,19,28,29,30,31,32,33] performed NSPT, which initially included oral hygiene instructions, encompassing brushing techniques and the use of interproximal hygiene devices, to control bacterial plaque. All patients underwent scaling and root planing (SRP) under local anesthesia at all locations with PPD ≥ 4 mm using ultrasonic and manual instrumentation with Gracey curettes.

Two studies [18,33] performed quadrant curettage in four visits over 21 days, scheduled on days 0, 7, 14, and 21. Two other studies [19,32] performed two sessions of SRP over two weeks. Vettore et al. [28] divided the SRP into four one-hour sessions, although they did not specify the interval between these sessions. For their part, Varadhan et al. [30] indicated that they completed the SRP in 7 days, while in two studies [29,31] they did not specify the duration of treatment.

Furthermore, the study by Bakri et al. [29] carried out a re-evaluation of SRP after 3 months, in sites where pockets larger than 4 mm persisted, further manual debridement was performed and those sites that showed satisfactory improvements in periodontal pocket reduction were included in supportive periodontal therapy (SPT). In another study [28] where SPT was performed, patients underwent SPT monthly until the completion of the 6-month study.

### 3.5. Results of Periodontal Parameters

The before and after NSPT results are detailed in Appendix A. Of the eight studies, four [24,26,27,34] showed improvements in CAL, PPD, and BOP at specific sites with periodontal pockets, and the other four studies [18,19,28,33] reported improvements in CAL, PPD, and BOP in the entire oral cavity. Among the studies that evaluated specific sites, there were four cohort studies [29,30,31,32], with follow-ups of 15 weeks [32], 3 months [30,31], and 6 months [29]. Of the studies that evaluated the entire oral cavity [18,19,28,33], three were cohort studies [18,19,33] with follow-ups of 6 months [19], 3 months [18], 6 weeks [33], and one was a case–control study [28], with a follow-up of 6 months.

Regarding the case–control study, Vettore et al. [28] reported results on mean PPD and CAL (%) and showed that stressed subjects did not show a reduction in the frequency of % PPD > 6 mm and CAL 4–6 mm and CAL > 6 mm, after 3 months of NSPT. In addition, showed a significant reduction in BOP and plaque index in all groups after NSPT.

Of the cohort studies, two studies [29,30] reported results with a mean of PPD and CAL (mm), showing that in stressed subjects, the mean PPD reduction was between 0.29 [30] and 1.4 mm [29], compared with 2.4 [30] and 2.32 [29] mm in non-stressed subjects. CAL gain was 0.15 mm [30] to 1.1mm [29] in the stressed subjects and 1.16 mm [30] to 2.3 mm [29] in the non-stressed subjects after 3 months [30] and 6 months [29] of NSPT. In addition, Dubar et al. [32] and Varadhan et al. [30] indicated that the reduction in PI and BOP was significantly greater in the non-stressed group compared to the stressed group.

Similarly, two other studies [19,31] showed that, in stressed patients, the average reduction in PPD ranged from 0.59 mm [19] to 1.74 mm [31]. Meanwhile, the gain in CAL was 0.60 mm [19] and 2.05 mm [31]. Petit et al. [19], at 6 months after periodontal treatment, presented a mean PPD reduction of 0.73 ± 0.11 mm and a decrease in periodontal pocket sites (PPD > 3 mm). Likewise, Ratika Lihala et al. [31] and Petit et al. [19] observed significant reductions in visible plaque and BOP after NSPT in stressed patients.

Bebars et al. [33], after 6 weeks of NSPT, demonstrated a mean PPD reduction of 0.9 mm in both groups, both in the group with lower levels of stress and in the group with higher levels of stress. The CAL gain was 0.7 mm for the patients with lower stress and 0.6 mm for the group with higher stress levels. In addition, Bebars et al. reported that at 6 weeks, the mean BOP level was 33% in patients with high stress and 20% in patients with stress. Romano et al. [18] divided the stress of the patients into high and low levels, finding a mean reduction of PPD (whole mouth) of 0.9 mm in the group with lower stress levels and 0.7 mm in the group with higher stress levels. Regarding CAL, patients with lower stress levels experienced a gain of 0.6 mm, compared to those with higher stress levels who had a gain of 0.3 mm. In addition, they found a greater reduction of the Full-Mouth Bleeding Score (FMBS) at 3 months in patients with lower stress (*p* < 0.001).

### 3.6. Control of Confounding Variables

Variables such as age, sex, smoking, race, and oral hygiene can influence psychological and periodontal variables. Therefore, it is crucial to adjust for these factors, which can be completed early in the research design or by statistical methods. Three studies [19,28,29] reported statistical adjustments for different combinations of variables, such as age, sex, and smoking (Table 2). Two other studies [18,32], although they did not specify statistical adjustments, indicated that matching and initial stratification of the groups were effective in controlling confounding factors. In addition, other studies mentioned that they used statistical methods, such as regression models, to adjust for these factors.

Different statistical approaches were employed to control for these variables. Five studies [18,19,28,29,32] used regression models, including univariate ANCOVA [28], multivariable ANCOVA [19], multiple regression, [29] logistic regression [32], and linear regression [18]. On the other hand, two studies [30,31] applied Pearson correlation analysis to explore the relationship between psychosocial factors and periodontal measures, and one study did not use correlation or regression analysis.

### 3.7. Odds Ratio

The studies that performed regression models presented their results as OR and 95% confidence interval (CI) to evaluate the effect of psychological stress on the response to NSPT (Table 2). Petit et al. [19] showed that increased DASS stress score was associated with worsening TPNC scores in terms of reduced BOP (OR = 1.02; *p* < 0.05) and mean PPD reduction (*p* < 0.05). It also indicated that negative coping strategies were also associated with a worsening of NSPT outcomes. Romano et al. [18], confirmed a significant effect of stress levels on full mouth bleeding score (FMBS), but not on mean PPD, as well as negative coping strategies, were associated with increased inflammation.

Bakri et al. [29] indicated that stress had a significant effect on changes in PPD CAL and elastase levels in periodontal pocket sites with bleeding using both methods (multiple imputation and last observation). Dubar et al. [32] presented an odds ratio (OR) of 0.64 for PPD and 2.82 for BOP, indicating that the stressed are 0.64 times more likely to experience PPD compared to the unstressed, although the difference is not significant. Additionally, an OR of 2.82 indicates that the highly stressed group is significantly more likely to have BOP compared to the non-stressed group.

### 3.8. Correlations

Two studies [30,31] used Pearson correlation statistical analysis to find a correlation between psychosocial factors and periodontal measures (Table 3). Varadhan et al. [30] found a weak positive correlation between stress and the presence of periodontal pockets of 4–6 mm and more than 8 mm, both at baseline and at the end of the third month. In addition, a weak positive correlation was observed between salivary cortisol levels and the level of clinical attachment loss (CAL), with losses of 1–2 mm in the control group and 5 mm or more in the stressed group.

Lihala et al. [31] established a statistically significant and positive correlation between the level of CgA in saliva and PI (*p* < 0.005), GI (*p* < 0.001) PD (*p* < 0.003), CAL (*p* < 0.001) GHQ (*p* < 0.030) in the AgP group. The correlation was highly significant for CAL and CgA levels in the CP and AgP group both at baseline and after NSPT.

### 3.9. Quality Assessment

The consensus on the quality assessments was uniform between the two reviewers. The average quality score of both reviewers for the eight studies analyzed was six, indicating high quality. The case–control study [28] obtained a perfect score of 9 points out of a total of 9. Regarding the cohort studies, one study [18] received a score of 9 points, another study [32] received a score of 8 points, and three studies [19,29,30] achieved a score of 7, demonstrating high methodological quality. The remaining two studies [31,33] achieved a score of 6, indicating moderate methodological quality (Table 4 and Table 5).

## 4. Discussion

The evidence from the included studies provides information on the relationship between psychological stress and periodontal treatment response. In three studies [28,29,30], it was observed that stressed individuals had a less favorable response to NSPT, compared to non-stressed. Bakri et al. [29] noted worse PPD and CAL scores at bleeding sites in patients with psychological stress, as assessed by the PSS-10 scale when compared to non-stressed patients. Varadhan et al. [30] demonstrated that the total percentage of affected areas with PPD was higher in the stressed group of patients (55.4%) compared to the non-stressed group (38%), and found a weak positive correlation between stress and the presence of PPD of 4–6 mm and more than 8 mm, both at baseline and the end of the third month. Vettore et al. [28] observed that after three months of NSPT, individuals with high levels of stress did not experience a significant reduction in the frequency of PPD greater than 6 mm compared to non-stressed patients. Further, Dubar et al. [32] did not observe significant differences after NSPT; however, they presented an OR of 0.64 for PPD and 2.82 for BOP, indicating that stressed individuals are more likely to experience PPD and BOP compared with non-stressed individuals.

Ratika et al. [31] reported that individuals with more severe periodontitis had major levels of non-baseline stress levels and, after periodontal treatment, a significant improvement in stress scores and periodontal parameters was observed. However, this improvement was greater in the group with lower stress levels at baseline. Petit et al. [19] highlighted a significant improvement in periodontal clinical parameters and also observed that increased depression and anxiety scores were associated with a poor response to periodontal treatment in terms of PPD and BOP reduction. Bebars et al. [33] found that the low-stress group achieved better control of gingival inflammation, compared with the medium- or high-stress group after the 6-week follow-up. Romano et al. [18] demonstrated that patients with lower stress levels showed superior results in terms of gingival inflammation, average probing depth, and number of deep periodontal pockets three months after periodontal treatment. They also confirmed a significant effect of stress levels on the result of full mouth bleeding on probing (FMBS).

Coping strategies alone had a strong influence on plaque index and bleeding on probing [18,33], but no differences were found about other clinical periodontal parameters at reassessment. It has been shown that psychosocial coping influences the results of periodontal treatment [18,19,33]. Individuals with inadequate coping have more severe periodontitis and tend to have a worse response to non-surgical periodontal treatment [19]. In contrast, individuals with adequate coping have mild periodontitis and a more favorable response to non-surgical periodontal treatment [18,33]. These results are similar to other studies that evaluated the role of stress and the influence of copying on periodontal disease [9,13,14,15] and its treatment [6,12].

Regarding biological biomarkers, four studies [19,29,30,32] evaluated cortisol levels, and observed that patients classified as stressed had higher cortisol levels than non-stressed patients [29], and Varadhan et al. [30] and Dubar et al. [32] showed a weak positive correlation between salivary cortisol levels and periodontal pocket depth. Dubar et al. [32] found that salivary cortisol levels were higher in the periodontitis patient group compared to the control group. There is evidence that in stressful situations, the autonomic nervous system that modulates the hypothalamic–pituitary–adrenal axis can release various substances such as cortisol [39] and increased salivary cortisol levels increase the progression of periodontitis [40]. This may be because salivary cortisol can modulate periodontal bacterial growth and the expression of virulence factors [39]. In addition, these results have been reinforced in previous systematic reviews, which indicate that salivary cortisol is increased in more aggressive or severe periodontitis [22,24].

Other biomarkers evaluated were CgA [19,31], elastase (ng/mL), and ICTP (ng/mL) [29] to assess the correlation between stress, type 1 collagen degradation, and bone loss caused by periodontitis, respectively. Lihala et al. [31] established a highly significant positive correlation in the group with higher periodontitis severity between saliva CgA level, periodontal clinical parameters (PI, PPD, CAL), and stress level (GHQ) before and after NSPT. Bakri et al. [29] observed that stress produced statistically significant changes in clinical measurements and elastase levels in gingival crevicular fluid (GCF) in deep pocket areas with bleeding and that the response to treatment was worse in the stressed group. They also observed that elastase levels in GCF were statistically higher in stressed patients, demonstrating that the greater the stress, the greater the degradation of type I collagen. These findings are consistent with previous studies [41,42] that have demonstrated a positive correlation between GCF elastase levels and periodontal probing depth (PPD) in patients with periodontitis. Regarding ICTP levels, although a decrease was observed at all sites after treatment, the effect of stress on this marker of bone turnover was not statistically significant. However, reductions in ICTP levels in the GCF were slightly greater in non-stressed patients at deep bleeding sites, suggesting a possible greater reduction in bone turnover at 6 months.

Of the included studies, five studies [18,19,28,29,32] employed different statistical methods to control for confounding variables (e.g., age, sex, smoking), two studies [30,31] made this adjustment in the initial phase of the research design, while on the other hand, two studies [30,31] applied correlation analysis to explore the relationship between psychological stress and periodontal measures. Noting that most of the studies (75%) indicated a positive relationship between stress and periodontal treatment response, another 12.5% observed a negative relationship between stress and response to NSPT, and the other 12.5% found some relationship characteristics described in the results of periodontal clinical parameters, verifying that most of the papers published to date examining this relationship have found positive associations on the influence of stress on periodontal treatment response.

The present systematic review has some limitations. First, the heterogeneity of the included studies in terms of design, study type, and follow-up time limited the possibility of performing a quantitative meta-analysis of the results. In addition, the different methods used to measure stress and periodontal parameters, as well as the differences in the populations studied, add complexity to interpreting the results. Despite the limitations, the evidence collected suggests that patients with higher levels of stress tend to have worse outcomes in periodontal parameters in the case–control study and cohort studies, with less reduction in probing depth and more bleeding on probing after NSPT. There is evidence that psychological stress can modulate inflammatory responses [19], alter angiogenesis [10], and reduce collagen production [29] and cell proliferation necessary for tissue repair, which can delay healing [43]. Further research is needed to understand these mechanisms and their clinical impact better.

These findings highlight the importance of assessing mental health at the time of diagnosis in order to optimize treatment. Adjunctive therapies to periodontal treatment could also be used, for example, the use of probiotics could be a possible complementary strategy in periodontal therapy for stressed patients. Systematic reviews [44,45] have highlighted the potential of probiotics to improve clinical outcomes in the treatment of periodontitis.

## 5. Conclusions

Based on the data obtained, we conclude that psychological stress has an influence on the effectiveness of the response to NSPT. Seventy-five percent of the studies analyzed found a positive correlation between high levels of stress and a less favorable response to periodontal treatment, evidenced by higher periodontal probing depth (PPD) and bleeding on probing (BOP) values. These findings suggest that stress may reduce the inflammatory response crucial for the elimination of pathogenic periodontal microorganisms after NSPT.

This highlights the importance of integrating stress management strategies into periodontal treatment, which could improve clinical outcomes in patients with periodontal disease. However, more longitudinal clinical studies are needed to understand these mechanisms.

## Figures and Tables

**Figure 1 jcm-14-01680-f001:**
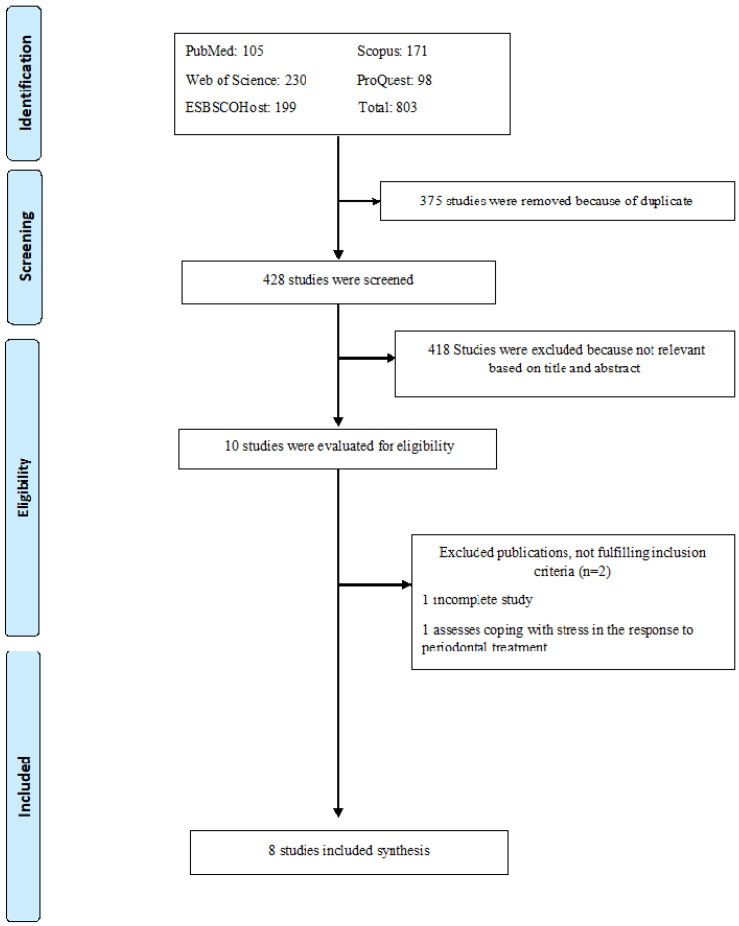
Flow chart.

**Table 1 jcm-14-01680-t001:** Characteristics of included studies.

Study and Year of Publication	Country	Study Design	Follow Up	Sample Size (Baseline)	Mean Age	Source of Funding
Vettore et al., 2005 [28]	Brazil	Case–control Study	6 months	N = 66 (male-45%; female-55%)	Age being over 35 years Mean age of the control group (PPD ≤ 4.0 mm; Stressed and non-stressed): 45.9 ± 8.3 The mean age of the test group 1 (PPD ≥ 4.0 and ≤6.0 mm): 46.4 ± 8.9 The mean age of test group 2 (PPD ≥ 6.0 mm): 46.1 ± 7.8	Not mentioned
Bakri et al., 2013 [29]	United Kingdom	Cohort study	6 months	N = 45 (16 male, 29 female)	Age > 20 years Mean age of the stressed group: 49.8 ± 9.7 Mean age of the non-stressed group: 44.6 ± 10.3	This study was partly self-funded with some funding from the Dental School, University of Sheffield
Varadhan et al., 2019 [30]	India	Cohort study	3 Months	N = 40 (40 male)	Age of 30 to 55 years old Mean age of the stressed group: Not mentioned Mean age of the non-stressed group: Not mentioned	Not mentioned
Ratika Lihala et al., 2019 [31]	India	Cohort study	3 months	N = 40	Age of 25 to 60 years old Mean age of the gingivitis group: 28.80 ± 2.78 Mean age of the Chronic Periodontitis group: 39.07 ± 4.79 Mean age of Aggressive Periodontitis group: 29.87 ± 4.50	NIL
Dubar et al., 2020 [32]	France	Cohort study	15 weeks	N = 60 (26 male; 34 female)	Age of 22 to 70 years old The median age of the periodontitis group (Stressed and non-stressed): 51 [38.0–65.0] The median age of the without periodontitis group (Stressed and non-stressed): 55.0 [22.0–70.0]	Concept Parodontal Association.
Bebars et al., 2021 [33]	Italy	Cohort study	6 weeks	N = 55 (20 male; 35 female)	Age of 20 to 80 years old Mean age of the high-stress group: 52.7 years Mean age of the low-stress group: 55.2 years	Not mentioned
Petit et al., 2021 [19]	France	Cohort study	6 months	N = 71 (31 male; 40 female)	Age of 29 to 74 years old Mean age of the group: 51.3 ± 10.1 years	Hopitaux Universitaires de Strasbourg through grant AAPJC 2013 HUS N° 5502
Romano et al., 2023 [18]	Italy	Cohort study	3 months	N = 90 (37 male, 53 female)	Age > 18 years Mean age of the high stress group: 57.1 ± 10.8 Mean age of the low stress group: 55.4 ± 10.7	Università degli Studi di Torino within the CRUI-CARE Agreement

**Table 2 jcm-14-01680-t002:** Study and year of publication, aim, statistical model and adjustment, variable, estimate (SD), OR (95% CI), outcome for periodontal disease and psychological factors, and main results, authors.

Study and Year of Publication	Aim of Analysis of Data	Statistical Model and Adjustment	Variable	Estimate (SD)	OR (95% CI)	*p*-Value	Outcome	Main Results
Vettore et al., 2005 [28]	Univariate ANCOVA was performed on the reduction of PPD and CAL frequencies > 4, 4–6, and >6mm with all psychosocial measures.	Univariate ANCOVA. Adjusted for dental plaque and number of cigarettes.	PI Number of cigarettes	No		0.249 0.898	Positive	The association between the reduction of deeper CAL frequencies (>6 mm) and scores of trait anxiety (TA) remained statistically significant after adjusting for dental plaque and number of cigarettes (*p* = 0.011).
Bakri et al., 2013 [29]	To study the change in clinical and biological data values over 6 months (6 months—baseline) comparing the patients who were stressed at baseline with those who were unstressed.	Multiple regression analyses. Adjusted for age, gender, and smoking	MI Changes in DNB sites	Estimated effect of stress (stressed–unstressed)			Positive	The effect of stress on changes in probing depths for DNB sites was statistically significant using the MI method. Still, it just failed to reach significance for the LOCF method for inputting missing values. The effect of stress on all the other parameters for DNB sites was not significant using both methods. In contrast, the effect of stress on all the changes for probing depths, clinical attachment levels, and elastase levels in GCF was statistically significant for deep bleeding sites using both methods. The effect of stress on changes in ICTP levels at deep bleeding sites was not statistically significant. Overall, both analysis methods for inputting missing data gave similar results for all measurements, other than for changes in probing depths at DNB sites.
			PPD	0.63	0.08 1.18	0.025 *		
			CAL	0.55	−0.10 1.20	0.096		
			Elastase level	1.708	−1.776 5.193	0.336		
			ICTP level	−0.090	−0.415 0.236	0.589		
			Changes in DB sites					
			PPD	0.93	0.50 1.36	<0.001 *		
			CAL	0.88	0.223 1.54	0.008 *		
			Elastase level	3.611	0.411 6.812	0.027 *		
			ICTP level	0.184	−0.124 0.493	0.238		
			LOCF Changes in DNB sites					
			PPD	0.54	−0.05 1.13	0.070		
			CAL	0.54	−0.16 1.23	0.125		
			Elastase level	2.17	−1.321 5.665	0.217		
			ICTP level	0.135	−0.431 0.702	0.632		
			Changes in DB sites					
			PPD	0.94	0.36 1.51	0.002 *		
			CAL	0.84	0.22 1.46	0.009 *		
			Elastase level	4.418	1.094 7.742	0.010 *		
			ICTP level	0.36	−0.073 0.810	0.99		
Dubar et al., 2020 [32]	To measure the effect of stress on the evolution of periodontal clinical parameters.	Logistic regression	Very Stressed Versus Non-Stressed/Managed Stress Patients				Negative	No significant association was found with the other periodontal parameters. After SRP, no association between periodontal parameters and psychosocial factors, both anxious and stress auto-questionnaires, was found.
			PI		Improvement 0.95 (0.31–2.85)	0.92		
					Degradation 1.48 (0.32–6.90)	0.62		
			PPD		Decrease 0.64 (0.13–3.16)	0.58		
					Increase 0.38 (0.02–6.35)	0.50		
			BOP		Disappearance 2.82 (0.84–9.51)	0.09		
					Emergence 1.03 (0.24–4.35)	0.97		
			Teeth mobility		Improvement 2.64 (0.22–30.97)	0.44		
					Degradation 2.64 (0.44–15.72)	0.29		
Petit et al., 2021 [19]	To determine the association between psychological status and SRP outcomes.	Multivariable analysis using variance components models	ΔPI	0.01 (0.00)	1.02 (1.00–1.04)	<0.001 *	Positive	Significant associations were demonstrated between SRP outcomes at 6 months and psychological scores. Indeed, the DASS-stress score was associated with worsened SRP outcomes regarding the evolution of BOP (OR = 1.02, *p* < 0.05) and mean PPD (*p* < 0.05) between baseline and 6 months. A specific focus has been made on the interpretation scores related to DASS. In this regard, a dichotomy has been made between patients with normal/mild (score: 0 to 18) vs. moderate/severe (score > 18) stress, normal/mild (score: 0 to 13) vs. moderate/severe (score > 13). The reduction of PPD > 3mm, mean CAL and CAL > 3mm were decreased in patients with moderate/high scores of DASS-stresses in comparison with normal/mild scores of patients (*p* < 0.05).
			ΔBOP			0.04 *		
			ΔPPD (mean)	0.01 (0.005)	1.01 (0.99–1.03)	0.03 *		
			ΔPPD > 3mm		1.0 (0.97–1.03)	0.31		
			ΔPPD > 5mm		0.94 (0.87- 0.01)	0.96		
			ΔPPD > 7mm			0.09		
		Adjusted for age, gender, and smoking	CAL (mean)	0.01 (0.008)	1.02 (1.00–1.04)	0.02 *		
			ΔCAL > 3mm		1.0 (0.98–1.02)	0.04 *		
			ΔCAL > 5mm		1.00 (0.97-1.03)	0.69		
			ΔCAL > 7mm			0.98		
Romano et al., 2023 [18]	Cohort	Linear regression models for final FMBS and PPD as dependent variables	Variable Mean FMBS at T1				Positive	A linear regression model considering both FMBS and PPD at the re-evaluation as dependent variables. Major stress level, avoidance coping strategy, number of PPD ≥ 6 mm at baseline, and FMPS at T1 were predictors of mean FMBS at the re-evaluation. On the other hand, major stress level, number of PPD ≥ 6 mm at baseline, and FMPS reduction were positive predictors of mean PPD at the re-evaluation, whereas coping strategy was not related.
			Stress level (major/minor)	15.855	11.477; 20.233	<0.001 *		
			Coping (approach/avoidance)	−5.160	−9.084; 1.236	0.011		
			N PPD ≥ 6 mm	0.284	0.028; 0.539	0.030		
			FMPS at T1	0.289	0.106; 0.472	0.002		
			Mean PPD at T1					
			Stress level (major/minor)	0.407	0.186; 0.628	<0.001 *		
			Coping (approach/avoidance)	0.121	−0.084; 0.325	0.244		
			N PPD ≥ 6 mm	0.011	0.006; 0.015	<0.001 *		
			ΔFMPS	−0.005	−0.009; −0.001	0.029		

ANCOVA, analysis of covariance; PI, plaque index; PPD, probing pocket deep; CAL, clinical attachment level; BOP, bleeding on probing; FMBS, full mouth bleeding score; FMPS, full mouth plaque score; DNB, deep non-bleeding; DB, deep bleeding; LOCF, last observation carried forward; MI, multiple imputations; ICTP, C-terminal teleopeptide of type collagen; SRP, scaling root planning; * *p* < 0.05.

**Table 3 jcm-14-01680-t003:** Statistical model and adjustment, OR, outcome for periodontal treatment and stress psychological, and main conclusion of the selected studies.

Study and Year of Publication	Aim	Statistical Model	Correlation Coefficient (R)	*p* Value	Outcome	Main Results, Authors’
Varadhan et al., 2019 [30]	To compare salivary cortisol level, pocket depths, and clinical attachment level	Pearson’s correlation coefficient (r) salivary cortisol level and pocket depths	Baseline Unstressed group	Baseline Unstressed group	Positive	A weak positive correlation between DSP scores and CAL was found in Group 1 of ≥5 mm at the baseline and end of the third month and between the salivary cortisol levels and pocket depths at the baseline and end of the third month, a weak positive correlation was found in the Group 2b of 4–6 mm, Group 1 and Group 2b of >8 mm. Weak positive correlation between the salivary cortisol levels and CAL found in the Group 1 of 1–2 mm, Group 2b of ≥5 mm.
			−0.122	0.608		
			3 months	3 months		
			−0.139	0.560		
			Baseline Stressed group	Baseline Stressed group		
			−0.267	0.256		
			3 months	3 months		
			−0.141	0.553		
		Pearson’s correlation coefficient (r) salivary cortisol level and clinical attachment levels	Baseline Unstressed group	Baseline Unstressed group		
			−0.047	0.843		
			3 months	3 months		
			−0.214	0.366		
			Baseline Stressed group	Baseline Stressed group		
			0.280	0.232		
			3 months	3 months		
			−0.018	0.940		
Ratika Lihala et al., 2019 [31]		Correlation of CgA with PI, GI, PPD, CAL, and GHQ scale		Baseline Correlation CgA and PI	Positive	The change in the levels of CgA post-intervention was statistically highly significant (*p* < 0.001) for all the 3 groups. Post-intervention, a statistically significant and positive correlation was established for salivary CgA and CAL (*p* < 0.034). Also, a statistically significant and positive correlation was established between salivary CgA level and PI (*p* < 0.005), GI (*p* < 0.001), PD (*p* < 0.003), CAL (*p* < 0.001) GHQ (*p* < 0.030) in AgP group.
				Gingivitis		
				0.953		
				CP		
				0.035 *		
				AgP		
				0.028 *		
				Correlation of CgA and GI		
				Gingivitis		
				0.378		
				CP		
				0.052		
				AgP		
				0.016 *		
				Correlation CgA and PPD		
				CP		
				0.015 *		
				AgP		
				0.001 **		
				Correlation CgA and CAL		
				CP		
				0.001 **		
				AgP		
				0.001 **		
				Correlation CgA and GHQ Scale		
				Gingivitis		
				0.136		
				CP		
				0.005 *		
				AgP		
				0.001 **		
				3 months		
				Correlation CgA and PI		
				Gingivitis		
				0.865		
				CP		
				0.660		
				AgP		
				0.005 *		
				Correlation of CgA and GI		
				Gingivitis		
				0.133		
				CP		
				0.278		
				AgP		
				0.003 *		
				Correlation CgA and PPD		
				CP		
				0.132		
				AgP		
				0.069		
				Correlation CgA and CAL		
				CP		
				0.034 *		
				AgP		
				0.001 **		
				Correlation CgA and GHQ Scale		
				Gingivitis		
				0.106		
				CP		
				0.088		
				AgP		
				0.030 **		

CgA, chromogranin A; CP, Chronic generalized periodontitis; AgP, aggressive generalized periodontitis; DSP, Derogatis Stress Profile; GHQ, The General Health Questionnaire; PPD, probing pocket deep; CAL, clinical attachment level; PI, plaque index; GI, gingival index; * Correlation is significant at the 0.05 level (two-tailed), ** Correlation is significant at the 0.001 level (two-tailed).

**Table 4 jcm-14-01680-t004:** Quality assessment for the case–control study using Newcastle–Ottawa scale (n = 1).

Study	Design	Selection (Score) R1	Selection (Score) R2	Comparability (Score) R1	Comparability (Score) R2	Outcome/Exposure (Score) R1	Outcome/Exposure (Score) R2	Total Score R1	Total Score R2
Vettore et al., 2005 [28]	Case–control	★★★★	★★★★	★★	★★	★★★	★★★	9	9

(1) High quality (total star score: 7 to 9/10); (2) moderate quality (total star score: 4 to 6); and (3) low quality (total star score: 0 to 3).

**Table 5 jcm-14-01680-t005:** Quality assessment for the cohort studies using Newcastle–Ottawa scale (n = 7).

Study	Design	Selection (Score) R1	Selection (Score) R2	Comparability (Score) R1	Comparability (Score) R2	Outcome/Exposure (Score) R1	Outcome/Exposure (Score) R2	Total Score R1	Total Score R2
Bakri et al., 2013 [29]	Cohort study	★★★	★★★	★	★	★★★	★★★	7	7
Ratika Lihala et al., 2019 [31]	Cohort study	★★★	★★★	★	★	★★	★★	6	6
Varadhan et al., 2019 [30]	Cohort study	★★★	★★★	★	★	★★★	★★★	7	7
Dubar et al., 2020 [32]	Cohort study	★★★★	★★★★	★★	★★	★★	★★	8	8
Bebars et al., 2021 [33]	Cohort study	★★★★	★★★★	★	★	★	★	6	6
Petit et al., 2021 [19]	Cohort study	★★★	★★★	★	★	★★★	★★★	7	7
Romano et al., 2023 [18]	Cohort study	★★★★	★★★★	★★	★★	★★★	★★★	9	9

(1) High quality (total star score: 7 to 9/10); (2) moderate quality (total star score: 4 to 6); and (3) low quality (total star score: 0 to 3).

## Data Availability

Data sharing does not apply to this article, as all the raw data on which this systematic review is based was extracted from studies previously published in the literature, which are duly cited in the text, figures and tables.

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
