# Peer review of "Psychological Stress Reduces the Effectiveness of Periodontal Treatment: A Systematic Review"

_jcm, 2025, doi:10.3390/jcm14051680_

Round 1

Reviewer 1 Report

Comments and Suggestions for Authors

The study on the influence of stress on periodontal disease pathophysiology and its impact on treatment outcomes is not a groundbreaking subject. However, this review provides a timely synthesis of evidence related to the topic, considering the growing interest in the impact of lifestyle on systemic wellbeing. The systematic review seems to be well-executed, while I have a few comments for the authors.

  1. A robust justification for conducting this systematic review is necessary, given the availability of existing reviews on this subject. In the introduction, please discuss and underscore previous reviews on this topic. In the discussion section, compare and contrast the findings (i.e., What is new?).
  2. The manuscript could benefit from a more comprehensive discussion on stress definitions and how stress affects immunity in the introduction section. This is the physiological basis for the investigation, and it would benefit new readers on the topic by the inclusion of representative references, such as PMID: 15250815, PMID: 34463997, etc.
  3. Is there a compelling reason for limiting the search to studies published after January 2000 and excluding those before 2000, as outlined in the author's published protocol? (https://pubmed.ncbi.nlm.nih.gov/39531258/)
  4. Did the authors explore the grey literature during their research?
  5. Have the authors considered including intervention trials in this review? These types of studies could provide evidence of causality which is of a higher level.
  6. More details on the data extraction procedure are required. (i.e., Was it conducted by one or two investigators? If so, how?)
  7. Please follow the PRISMA flow diagram template for reporting; templates are available. Refer to the PRISMA 2020 flow diagram — PRISMA statement. The resolution of the figures needs to be improved.
  8. Please revise the manuscript for minor grammatical errors and improve the overall language."
Comments on the Quality of English Language

Please find comments above for authors. 

Reviewer 2 Report

Comments and Suggestions for Authors

Dear Authors,

I reviewed the manuscript "Psychological Stress Reduces the Effectiveness
of Periodontal Treatment: A Systematic Review." The topic addressed in your
paper is very interesting, and below I report some doubts/suggestions for
each section.

Keywords:
Please remove the numbers 1,2,3,4 after the words. Additionally, it is
recommended to include between 5 and 10 keywords.

Introduction:
The introduction should more clearly integrate the current evidence on the
relationship between stress and periodontal health. Specifically, in 2024,
a Cross-Sectional Study on Periodontal Health related to Psychological
Stress was published, which could be of interest to the reader.

Additionally, describe in which other fields of medicine a similar review
has been conducted (for example, studies evaluating stress and treatment in
oncology have already been performed).

Materials and Methods:
This section is well-described and well-structured.

Results:
The section "Search results and excluded trials" should be moved to the
Materials and Methods section.
Figure 1. Flow Chart is not legible. The image quality needs to be improved.
Table 1 is confusing. It is recommended to create a separate table
specifically for the data in the column "Sample size (baseline)."
Table 2 is too long. It is advisable to summarize all the sentences
included in the table.
Table 5 contains the following text: "(1) alta calidad (puntuación total: 7
a 9/10); (2) calidad moderada (puntuación total: 4 a 6); y (3) baja calidad
(puntuación total: 0 a 3)." The entire text must be in English.
Conclusions:
The scientific relevance of the conclusions is missing. This section can be
improved considering the amount of data presented in the study.

Reviewer 3 Report

Comments and Suggestions for Authors

Dear Authors,

I carefully read your manuscript entitled: "Psychological Stress Reduces the Effectiveness of Periodontal Treatment: A Systematic Review".

Here my comments.

INTRODUCTION

1. While the paragraph mentions that stress is associated with immune dysfunction and inflammatory processes, the causal relationships between stress and periodontitis are not clearly defined. It would benefit from a more in-depth explanation of how these interactions unfold at a mechanistic level.

2. The paragraph references a number of studies without delving into their methodologies or sample sizes. Phrases like: "some studies" line 40, "recent studies" line 45, are vague and lack specificity, which weakens the credibility of the claims. Citing specific, well-designed studies could strengthen the argument.

3. While stress is highlighted as a contributing factor to periodontitis, the paragraph does not fully explore other potential psychological or environmental influences that may interact with stress in contributing to periodontal disease. For example, socio-economic factors or genetics are not addressed.

4. At line 65, you write: "no systematic review has been conducted to specifically analyze the effect of psychological stress on periodontal treatment outcomes". However, you cited your own systematic review with references n. 18 as an ongoing protocol. I think it is controversial. Please explain it.

Matherial and methods, Results sections are clear and well-organized.

DISCUSSION

I appreciated you cited the limitations of this review. However, I think that a brief paragraph should be added regarding other auxiliary procedures to the gold-standard therapy, such as the use of some strains in Probiotics, please refer to https://pubmed.ncbi.nlm.nih.gov/38668014/.

Round 2

Reviewer 2 Report

Comments and Suggestions for Authors

Congratulations. The paper is worthy of publication in its present form

Reviewer 3 Report

Comments and Suggestions for Authors

Dear Authors, I appreciated the modifications done. I suggest the acceptance in the present form.